# Simultaneous Quantitation of S(+)- and R(−)-Baclofen and Its Metabolite in Human Plasma and Cerebrospinal Fluid using LC–APCI–MS/MS: An Application for Clinical Studies

**DOI:** 10.3390/molecules25020250

**Published:** 2020-01-08

**Authors:** Qingfeng He, Yashpal S. Chhonker, Matthew J. McLaughlin, Daryl J. Murry

**Affiliations:** 1Clinical Pharmacology Laboratory, Department of Pharmacy Practice and Science, University of Nebraska Medical Center, Omaha, NE 68198, USA; qingfeng.he@unmc.edu (Q.H.); y.chhonker@unmc.edu (Y.S.C.); 2Division of Rehabilitation Medicine, Children’s Mercy Kansas City, UMKC School of Medicine, Kansas City, MO 64108, USA; mjmclaughlin@cmh.edu; 3Fred and Pamela Buffett Cancer Center, University of Nebraska Medical Center, Omaha, NE 68198, USA

**Keywords:** LC-MS/MS, baclofen, 3-(4-chlorophenyl)-4 hydroxybutyric acid (CHBA), chiral separation, CSF, spasticity, pharmacokinetics

## Abstract

Baclofen is a racemic mixture that is commonly used for the treatment for spasticity. However, the optimal dose and dosing interval to achieve effective cerebral spinal fluid (CSF) concentrations of baclofen are not known. Moreover, it is unclear if there are differences in the ability of R- or S-baclofen to cross the blood–brain barrier and achieve effective CSF concentrations. We have validated a liquid chromatography coupled with tandem mass spectrometry (LC-MS/MS) method with improved selectivity and sensitivity for the simultaneous quantitation of R- and S-baclofen and metabolites in plasma and CSF. Protein precipitation by acetonitrile was utilized to obtain an acceptable recovery of the analytes. The detection and separation of analytes was achieved on a 48 °C-heated Crownpak CR(+) column (150 mm × 4.0 mm, 5μ) with elution using 0.4% formic acid (FA) in water and 0.4% FA in acetonitrile as the mobile phase running at a flow rate of 1.0 mL/min. Accurate quantitation was assured by using this MS/MS method with atmospheric pressure chemical ionization in multiple reaction monitoring (MRM) mode. Therefore, this method is enantioselective, accurate, precise, sensitive, reliable, and linear from 1 to 1500 ng/mL for baclofen and 2 to 4000 ng/mL for the metabolites. An additional method was developed to separate racemic baclofen 3-(4-chlorophenyl)-4 hydroxybutyric acid metabolites for individual concentration determination. Both validated methods were successfully applied to a clinical pharmacokinetic human plasma and CSF study evaluating the disposition of baclofen and metabolites.

## 1. Introduction

Baclofen (4-amino-3-*p*-chlorophenyl butyric acid) (Figure 1) is a racemic mixture, comprising equal amounts of the R- and S-enantiomers and was initially designed as a treatment for seizures in the late 1960s. Baclofen is now the most commonly prescribed oral medication to treat spasticity of central origin [1,2], such as cerebral palsy [3], hemi- and tetraplegia [4], and multiple sclerosis [5]. Despite the widespread use of baclofen, substantial variability exists in exposure and clinical response even when patients receive similar doses of baclofen [6,7].

Clinically, racemic baclofen is often given orally with a reported bioavailability of 70% to 80% [8,9]. The bioavailability of baclofen has been reported to decrease with increasing doses, suggesting saturable absorption [10]. Approximately 30% of the parent drug is bound to albumin [8], and peak plasma concentrations are obtained, on average, within 2 h of administration [11]. Baclofen is extensively distributed throughout the body, including distribution into the cerebrospinal fluid (CSF) [4,8,11]. The parent drug is primarily excreted unchanged by glomerular filtration with one major pharmacologically inactive metabolite: 3-(4-chlorophenyl)-4-hydroxybutyric acid (CHBA) [8,12]. The elimination half-life has been reported to range from 2 to 6 h [8,9]. However, elimination can be delayed when renal excretion is saturated following high doses with studies reporting a half-life up to 34.6 h [13]. Historically, pharmacokinetic studies have not differentiated the amount of R- and S-baclofen during analysis; however, it is known that R-baclofen is significantly more potent than S-baclofen, and there is clear stereoselectivity at the GABA-B receptor site for R-baclofen. The IC_50_ of R-baclofen is 130 times less than S-baclofen when administered independently [14]. On the other hand, the intrathecal administration of baclofen using an implanted pump has been widely adapted in clinical settings [15]. Lower doses of baclofen can be given via the intrathecal route to patients to gain more effective therapeutic outcomes with fewer adverse effects compared with oral administration [16]. However, this technique makes it more difficult to detect and evaluate baclofen pharmacokinetics in human plasma and CSF, considering that the recommended starting dose is as low as 100 µg per day [17]. Although the proposed mechanism for transport across the blood–brain barrier (BBB) is a large neutral amino acid transporter (LAT1) [18], stereoselectivity at this transporter for either of the two enantiomers of baclofen or its primary metabolite could cause significant downstream effects. Given the differences in IC_50_, the potential for stereoselectivity at transporters involved in the disposition of baclofen, and the low dose of baclofen administered intrathecally, there is a need to develop a sensitive analytical method to determine which enantiomers and metabolites cross the BBB to improve drug therapy and limit systemic toxicities.

Baclofen has been determined in a variety of different biological matrices using analytical techniques based on gas chromatography [19,20], liquid chromatography-mass spectrometry (LC-MS) [21,22], as well as electrophoresis [23]. However, the methods mentioned above require complex procedures to prepare samples, including using solid phase extraction (SPE). In this manuscript, we describe two separate methods in human plasma and CSF; the first being a rapid and sensitive method to determine the concentration of R- and S-baclofen enantiomers and the second method to separate and quantitate the R- and S-CHBA enantiomers (Appendix A). The development of this simple and rapid sample preparation combined with optimized analytical methods will be beneficial for monitoring baclofen systemic exposure and the relation with effect and toxicity.

## 2. Results and Discussion

### 2.1. Chromatographic and Mass Spectrometric Optimization

Positive and negative atmospheric pressure chemical ionization (APCI) and electrospray ionization (ESI) conditions were tested for optimized MS conditions to detect baclofen and its metabolites. The signal intensity of baclofen and CHBA metabolites were the highest using the APCI probe in positive and negative mode, respectively. The analytical separation for R and S-baclofen was performed using a Crownpak CR(+) (4 mm × 150 mm, 5μ) column, as previously described [24]. However, this method was not able to separate the metabolites S-CHBA and R-CHBA, which co-eluted together under these conditions. A separate method was developed utilizing a Chiralcel OJ R-RH (2.1 mm × 150 mm, 5μ) column to separate the R- and S-CHBA metabolites. Moreover, the metabolites were more readily detected utilizing the APCI probe operated in negative mode (Appendix A). During method optimization, the mass spectra for baclofen and baclofen-*d*_4_ (IS) revealed precursor ion peaks at *m*/*z* 214.10, and 218.10, respectively in positive APCI mode. The CHBA metabolite revealed precursor ion peaks at *m*/*z* 213.15 in negative mode APCI. The fragmentation of analytes and IS were optimized automatically using the auto-optimize function of the LabSolutions software and a stock solution of each compound. The most abundant parent > product ion with the highest sensitivity for baclofen, baclofen-*d*_4_, and CHBA metabolite were *m*/*z* 214.10 > 151.10 (APCI+), 218.15 > 155.10(APCI+), and 213.20 > 151.20 (APCI−) (Figure 2), respectively. The final parameters are shown in Table 1.

R- and S-baclofen were separated and quantitated using the final mobile phase consisting of 0.4% formic acid (FA) in water and 0.4% FA in acetonitrile (ACN)utilizing a Crownpak CR(+) column. The total flow rate was 1.0 mL/min, and the column temperature was set to 48 °C, which produced the best peak shape and no endogenous interference for the R- and S-baclofen enantiomers.

R and S-CHBA metabolites were separated using a Chiralcel OJ R-RH column with 0.4% FA in water and 0.4% FA in ACN (86:16, *v*/*v*) as the mobile phase and a flow rate of 0.1 mL/min. These conditions resulted in acceptable peak shape for R and S-CHBA metabolites with no interference of endogenous compounds at the retention time for both baclofen and its CHBA metabolites (Appendix A). Post-column infusion of 0.5% ammonium hydroxide (NH_4_OH) in water–ACN (80:20, *v*/*v*) was accomplished using a three-way connector before mass detection to improve ionization for the R- and S-CHBA metabolites, which utilized negative APCI mode ionization (data not shown) [2].

The retention times for S-baclofen, R-baclofen, S-CHBA, and R-CHBA were 3.5, 5.4, 21.0, and 23.5 min in both assays, respectively.

Previous studies extracting baclofen from biomatrices utilized different extraction methods including solid phase extraction (SPE) [2,21,24], liquid–liquid extraction [25], and protein precipitation extraction (PPE) [2]. However, SPE [24] in our hands resulted in poor CHBA recovery, with only approximately 5% of the metabolite recovered (data not shown). In order to simplify procedures and improve metabolite recovery, we utilized a PPE method to achieve approximately 90% absolute recovery for baclofen and CHBA metabolites (Section 2.2.6). The addition of a simple and inexpensive sample extraction resulted in exhanced recovery and no matrix effect. The human plasma and CSF mixed with blank plasma both resulted in similar recovery and matrix effect. The human plasma was used to construct CSF calibration curves and quality controls (QC)s, whereas a CSF real sample was mixed with 100 uL blank plasma to nullify the potential matrix effects. The same PPE extraction method was utilized for both assays. The method is quantitative, validated for both plasma and CSF samples, and linear from 1 to 1500 ng/mL for baclofen. Our validated method is accurate, precise, and sensitive, allowing for the routine analysis of baclofen and metabolites in clinical samples [26].

### 2.2. Assay Validation

#### 2.2.1. Sensitivity

Sensitivity was defined as the lower limit of quantitation (LLOQ), where the signal-to-noise (S/N) response of all analytes was equal or greater than 10-fold compared to the blank response. The lowest concentration for quantitation in assay with the percent Relative Standard Deviation (%RSD) <20% was taken as LLOQ and was found to be 1.0 ng/mL for R-baclofen and S-baclofen and 2.0 ng/mL for CHBA (Racemic). In the second assay, the LLOQ was found to be 1.0 ng/mL for R-CHBA and S-CHBA and 2.0 ng/mL for baclofen (Racemic).

#### 2.2.2. Specificity and Selectivity

Assay specificity and selectivity was evaluated by analyzing blank human plasma samples to investigate possible interferences at the retention time of interest (Figure 3). No co-eluting peaks greater than 20% of the analyte area at the LLOQ level or greater than 5% of the area of internal standard (IS) were observed (data not shown).

#### 2.2.3. Calibration Curve and Linearity

To calculate concentrations for unknown samples, linear regression with and without intercepts (*y* = *mx* + *c* and *y* = *mx*) and 1/*y*^2^ weighting was utilized. The final assay was linear over the concentration range of 1 to 2000 ng/mL for R- and S-baclofen and from 2 to 4000 ng/mL for racemic CHBA. In the second assay, concentrations ranging from 1 to 2000 ng/mL for R- and S-CHBA and 2 to 4000 ng/mL for racemic baclofen were linear.

#### 2.2.4. Carry-Over

There was no significant carry-over effect based on the response of a blank sample injected following a spiked sample (data not shown).

#### 2.2.5. Accuracy and Precision

The inter-day and intra-day accuracy (% bias) and precision (%RSD) results for the detection of baclofen and CHBA metabolites in human plasma samples at the lower limit of quantification (LLOQ), low-quality control (LQC), middle-quality control (MQC), and high-quality control (HQC) are presented in Table 2. Using a Crownpak column to separate R-and S-baclofen and total CHBA metabolites, the RSD of inter- and intra-day accuracy values were between −5.2 and 18.4%, and inter- and intra-day precision values ranged from 1.1 to 16.6% (Table 2).

Using a Chiralcel column to separate R- and S-CHBA metabolites and total baclofen, the RSD of inter-day precision values were between 3.3 and 18.7% with intra-day precision values ranging from 1.7 to 9.7%, which indicated acceptable assay precision (Appendix A).

#### 2.2.6. Recovery and Matrix Effect

The absolute recoveries were calculated at the LQC, MQC, and HQC concentration as the mean peak area of an analyte spiked before extraction to the mean peak area of an analyte spiked post-extraction multiplied by 100. The absolute mean recovery results for S-baclofen, R-baclofen, and CHBA (racemic) respectively are provided in Table 3. The recovery of IS was found to be 88.1 ± 6.1% and 86.4 ± 6.4% for S- and R-baclofen-*d*_4_, respectively. In general, baclofen and its metabolite recovery was approximately 90%.

Plasma calibration curve (CC) and QCs to were found to mimic human CSF mixed with 100-µL blank plasma with similar recovery and matrix effects using analyte/IS peak area ratios. The matrix effect for all analytes at LQC, MQC, and HQC concentration levels in plasma and CSF were < ±15% (85.2–105.2%). The IS normalized matrix effect for all analytes were < ±15% (91.9–113.6). The recoveries of all analytes in the plasma were similar to those in CSF (Table 3), indicating a no-matrix effect for the study samples. Our result confirms that plasma CC and QCs mimic human CSF mixed with 100-µL blank plasma, as they have shown similar recovery and matrix effects using analyte/IS peak area ratios.

#### 2.2.7. Stability

R and S-baclofen and racemic CHBA were found to be stable under the conditions tested. The stability data are provided in Table 4. Overall, in the different stability studies, concentrations of analytes remained within 15% of the actual corresponding LQC, MQC, and HQC concentrations.

#### 2.2.8. Application of the Method for Clinical Sample Analysis

Therapeutic concentrations of baclofen are considered to be 80–400 ng/mL in plasma [27]. Baclofen oral doses between 30 to 90 mg daily are related to total baclofen plasma concentrations of 68 to 650 ng/mL [28]. The validated LC-MS/MS method was utilized to determine the concentration of S-baclofen, R-baclofen, and its CHBA metabolite in plasma and CSF following single oral baclofen doses of 15–30 mg. The plasma concentrations of R- and S-baclofen (median: range) were detected during therapy: 54.8 (6.4–334.0) ng/mL and 59.7 (5.6–246.4) ng/mL, respectively. The CSF concentrations of R- and S-baclofen detected during therapy ranges were 1.7–20.8 ng/mL and 1.4–14.4 ng/mL, respectively. The ratio of plasma to CSF for R-baclofen and S-baclofen ranges were 3.5–35.2 and 4.0–33.9, respectively. The R-CHBA metabolite was not detected in plasma or CSF samples (Appendix A), while the S-CHBA metabolite ranges were 8.8–48.6 ng/mL and 1.3–15.4 ng/mL in plasma and CSF, respectively.

The present bioanalytical method is useful for the therapeutic monitoring of baclofen therapy and evaluating strategies to individualize baclofen therapy. Drowsiness is a common side effect of baclofen therapy and is suggested to be dose-dependent. The evaluation of baclofen and metabolite concentrations along with assessments of efficacy and toxicity will be important to further improve the use of baclofen. The overall goal of treatment is to individualize baclofen therapy to improve the therapeutic response while minimizing systemic toxicities. While the study presented here provides a useful tool to monitor baclofen therapy, the limited number of subjects limits our conclusions. Additional studies with a larger number of patients will be crucial to the development of individualized baclofen therapy.

## 3. Materials and Methods

### 3.1. Chemicals and Materials

S-baclofen, R-baclofen, S-CHBA, R-CHBA, and baclofen-*d*_4_ of pharmaceutical grade were obtained from Toronto Research Chemicals (Toronto, Ontario, Canada). Mobile phase solutions were purchased from Fisher Scientific (Fair Lawn, NJ, USA) including HPLC-grade methanol (MeOH), acetonitrile (ACN), and formic acid (FA). Centrifuge tube filters were obtained from Corning Co. (Corning, NY, USA). A water purification system (ThermoFisher Scientific, Grand Island, NY, USA) was utilized for ultrapure water. Other reagents were purchased from standard chemical suppliers and were of analytical grade or higher.

### 3.2. Liquid Chromatographic and Mass Spectrometric (LC/MS) Conditions

A Shimadzu Nexera Ultra high-performance liquid chromatography (UHPLC) system (Shimadzu Scientific Instruments, Columbia, MD) was used for chromatographic separation and included a binary pump system (LC-30 AD), an auto-sampler (SIL-30AC) and a column oven (CTO-30AS). Mass spectrometric detection was performed using a LC-MS/MS 8060 system (Shimadzu Scientific Instruments, Columbia, MD, USA) and an atmospheric pressure chemical ionization (APCI) probe.

Two methods were validated to separate and quantitate all four compounds of interest. One method to separate both R-and S-isomers for baclofen and racemic CHBA metabolite (both R and S isomers) and a second method to quantitate R-CHBA and S-CHBA and racemic baclofen.

R- and S-baclofen separation and racemic CHBA assay: All chromatographic separations were performed with a Crownpak CR(+) column (4 mm × 150 mm, 5μ; Daicel Chemical Industries, Ltd., Osaka, Japan) equipped with a guard column (Crownpak CR(+) 4.00 × 10 mm, 5μ). The system was calibrated in isocratic mode with a mobile phase consisting of 0.4% FA in water-0.4% FA in acetonitrile (ACN) (84:16, *v*/*v*) at a flow rate of 1.0 mL/min along with a column oven heated to 48 °C. The total run time was 11 min. The injection volume of all samples was 20 μL.

Post-column infusion of 0.5% ammonium hydroxide (NH_4_OH) in water–ACN (80:20, *v*/*v*) at 0.1 mL/min was performed using a three-way connector before mass detection. Detailed parameters about each method and comparison are shown in Table 1.

The MS parameters were optimized using the auto-optimization method as implemented in LabSolutions software Version 5.6 (Shimadzu Scientific, Inc., Columbia, MD, USA). The final parameters for each analyte and IS are shown in Table 1.

### 3.3. Preparation of Stock, Calibration Standards, and Quality Control Samples

Stock solutions of R- and S-baclofen, R-CHBA, and S-CHBA were prepared by dissolving 1 mg of each compound into 1 mL of water. The stock solutions were transferred to an Eppendorf vial and stored at −20 °C for future use.

For the quantitation of R- and S-baclofen and R-CHBA, and S-CHBA, these four stock solutions were further diluted with 50% methanol in water to obtain a concentration of 20 μg/mL, which was then further diluted respectively for the preparation of working standard solutions. The calibration standards (CCs) were individually prepared by spiking 20 µL of mixed working standard solution into 200 µL of plasma to obtain a concentration ranging from 1 to 2000 ng/mL (1, 2, 5, 10, 50, 200, 1000, and 2000 ng/mL) for R- and S-baclofen. For racemic CHBA, the concentration ranged from 2 to 4000 ng/mL (2, 4, 10, 20, 100, 400, 2000, and 4000 ng/mL). Apart from CCs, five replicates of each QC concentration were also prepared at the same time. QC samples were prepared at 1 ng/mL (lower limit of quantification, LLOQ), 5 ng/mL (low-quality control, LQC), 500 ng/mL (middle-quality control, MQC), and 1500 ng/mL (high-quality control, HQC) for R- and S-baclofen. The corresponding concentration for CHBA was doubled due to its racemic mixture. Plasma was used to construct CSF calibration curves and QCs, whereas CSF samples were mixed with 100 uL of blank plasma to nullify any potential matrix effect.

For the internal standard, 1 mg/mL stock solution of baclofen-*d*_4_ (racemic) was prepared by dissolving 1 mg in 1 mL of water. The sample was transferred to an Eppendorf vial and stored at −20 °C for future use. When preparing working IS stock solution, the baclofen-*d*_4_ was diluted with 50% methanol to make a 5 μg/mL solution.

### 3.4. Plasma and CSF Sample Preparation

The extraction of baclofen and metabolites was performed using a simple protein precipitation extraction (PPE) method. Briefly, 200 µL of blank human plasma, CSF, QC, or study sample was transferred into a 1.5-mL polypropylene (PP) tube, and CCs were spiked with 20 µL of working stock solution as well as 20 µL of IS working solution (baclofen-*d*_4_, 5 μg/mL). Ice-cold ACN (1 mL) was added to each sample, and the solution was mixed on a vortex mixer for 30 s. Subsequently, 100 µL of 0.1% formic acid in water was added, and the mixture was vortexed for 2 min and then centrifuged at 18,000× *g* for 10 min at 4 °C. The supernatant (approximately 950 µL) was transferred to a 13 × 100 mm glass tube and evaporated under nitrogen at 40 °C. The dried sample was reconstituted with 100 μL of 0.1% FA–MeOH (85:15, *v*/*v*), vortexed for 30 s, and centrifuged again at 18,000× *g* for 15 min at 4 °C, after which the supernatant (approximately 80 uL) was transferred to an autosampler vial.

### 3.5. Assay Validation

The developed LC-MS/MS assay for R- and S-baclofen separation and racemic CHBA was fully validated in plasma, and the assay for R- and S-CHBA separation was validated for sensitivity, selectivity, accuracy, and precision (Appendix A) in plasma, according to the current FDA guidelines [29].

The assay sensitivity was evaluated by calculating the accuracy and precision (A&P) at an assigned LLOQ with ±20% of nominal concentration. Additionally, sensitivity was determined via the signal-to-noise ratio (S/N) of the analyte response in the calibration standards, which was required to be greater than three for the LOD and 10 for the LLOQ.

Assay selectivity and specificity were determined by comparing the results of six different blank human plasma samples with those of R-baclofen, S-baclofen, R-CHBA, S-CHBA, and the IS-spiked sample for the assessment of potential interferences with endogenous substances at the retention time of analytes or IS.

The peak area ratio (analyte/IS) versus concentration for all analytes was plotted to generate calibration curves for each analyte. The calibration curve consisted of 12 data points, including a blank sample (no analyte or IS), a zero calibrator (blank + IS), and 10 CCs. All standard concentrations were required to be within ±15% standard deviation (SD) from the nominal value with the exception at LLOQ, which was set at ±20%.

The carry-over effect was determined by injecting two blank samples (no analyte or IS) following an HQC sample.

The method was validated for intra-day accuracy and precision by the analysis of five replicates of the QC samples injected on the same day. To assess inter-day accuracy and precision, five samples at each QC concentration were injected daily for three separate days. Precision was defined as the percent relative standard deviation (%RSD) with an acceptance criteria of ±15% (except for ±20% at LLOQ). Accuracy was defined as the percent bias (% bias) with the same acceptance criteria as precision.

### 3.6. Recovery and Matrix Effect

The peak area of R and S-baclofen, IS, and racemic metabolites in extracted samples (plasma and CSF mixed with blank plasma, 1:1) at three different QC concentrations (i.e., LQC, MQC, and HQC) were compared to the peak area or peak area ratio obtained for equivalent concentrations for each analyte in post-extracted samples..

To assess the effect of the matrix on analyte response, blank human plasma and CSF mixed with blank plasma were processed, and the dry post-extract was spiked with analyte and prepared equivalent to the QCs. The average of peak area from all the analytes spiked in the blank matrix was compared to the response for QCs prepared in reconstitution solvent. The absolute matrix effect (ME) and IS normalized ME were calculated as described in Equations (1) and (2).
(1)$$ME=\frac{Mean\;Peak\;area\;of\;analyte\;\mathrm{spiked}\;\mathrm{post}\text{-}\mathrm{extraction}}{Mean\;Peak\;area\;of\;analyte\;in\;Solvent}\times 100$$
(2)$$IS\;normalized\;ME=\frac{Mean\;Peak\;area\;ratio\;of\;analyte/IS\;\mathrm{spiked}\;\mathrm{post}\text{-}\mathrm{extraction}}{Mean\;Peak\;area\;ratio\;of\;analyte/IS\;in\;Solvent}\times 100$$

### 3.7. Stability

Baclofen and CHBA metabolite stability in human plasma samples was determined for bench-top storage (20 °C for up to 6 h), three freeze–thaw cycles (room temperature to −80 °C to room temperature), long-term stability (−80 °C for 40 days) before and after processing the samples, and auto-sampler stability (4 °C for 72 h). The sample concentration for all analytes under each individual condition was tested, and the mean values for accuracy and precision were calculated.

### 3.8. Clinical Study and Therapeutic Drug Monitoring

Biospecimens were obtained from patients (<20 years of either sex) presenting with spastic or dystonic cerebral palsy who were having their CSF space accessed as part of routine clinical care at the Department of Rehabilitation Medicine at Children’s Mercy-Kansas City. All specimens were obtained with informed permission/assent. The protocol was approved by the institutional ethics committee (approval number 17120739). Each parent or legal guardian provided permission for their child to participate in this study and were provided assent for participation. Both the parent/guardian and the participant were informed of the objectives, nature, and potential risks of the study. Participants received their current oral dose of baclofen, ranging from 15 to 30 mg per dose. Blood samples (2 mL in 3.8% tri-Sodium Citrate 9:1, *v*/*v*) were obtained at the same time as CSF samples (±5 min). The blood samples were centrifuged at 2000× *g* at 4 °C for 10 min to separate the plasma. All plasma and CSF samples were stored in sealed tubes at −80 °C until analysis. Two participants provided paired clinical specimens and were evaluated as part of method development.

## 4. Conclusions

We have developed two reproducible and efficient LC-MS/MS methods for separating and quantitating racemic baclofen and its racemic CHBA metabolites, respectively. Compared to previous studies, we have improved the recovery of the metabolites during sample processing. In addition, our results confirmed that there is no R-CHBA metabolite formed in the CSF or that crosses from the plasma to CSF. Therefore, a single method to determine racemic baclofen and the CHBA metabolite was also developed during this study and is sufficient to monitor the S-CHBA metabolite in the CSF, as no R-CHBA was observed in CSF samples. Baclofen plasma-to-CSF concentration ratios were also determined to be approximately 3.8–34.6, which could be beneficial to evaluate therapeutic effect without performing an invasive lumbar puncture. This method will help to understand the biodistribution and pharmacodynamic effects of baclofen and metabolites, potentially improving the efficacy and safety of baclofen therapy.

## Figures and Tables

**Figure 1 molecules-25-00250-f001:**
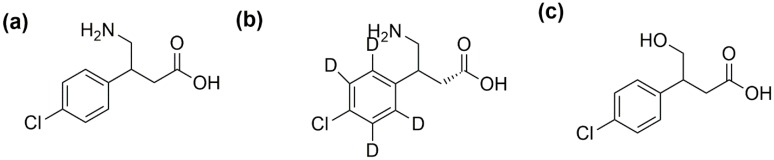
The chemical structure of (**a**) baclofen, (**b**) baclofen-d4, and (**c**) 3-(4-chlorophenyl)-4 hydroxybutyric acid (CHBA) metabolite.

**Figure 2 molecules-25-00250-f002:**
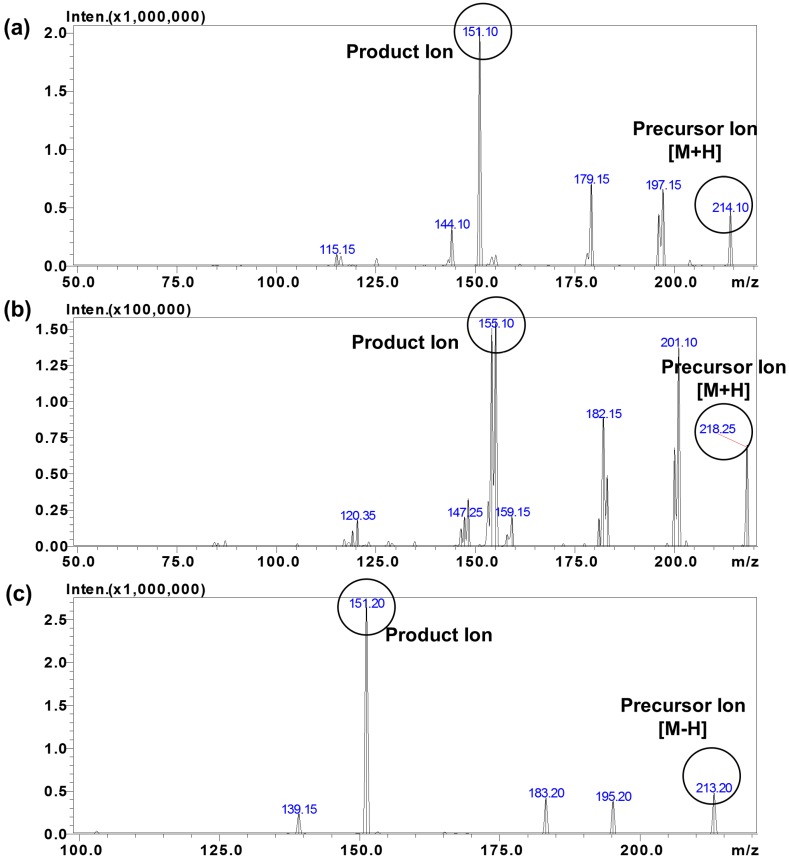
MS product ion spectra of (**a**) baclofen, (**b**) baclofen-d4, and (**c**) CHBA metabolite in +/− atmospheric pressure chemical ionization (APCI) mode.

**Figure 3 molecules-25-00250-f003:**
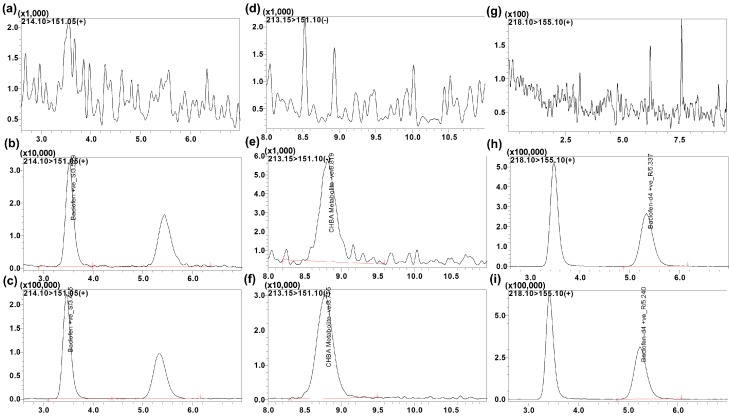
Representative multiple reaction monitoring (MRM) ion chromatograms of (**a**) blank plasma using the conditions for baclofen, (**b**) plasma spiked with baclofen showing the two isomers (S-isomer 3.5 min and R-isomer 5.4 min at 5 ng/mL), (**c**) extracted human plasma sample following baclofen administration showing baclofen isomers, (**d**) blank plasma using the conditions for CHBA metabolite, (**e**) plasma spiked with CHBA metabolite showing the racemic peak for both R- and S-isomers (not separated, 8.7 min, at 5 ng/mL), (**f**) human plasma sample following baclofen administration showing racemic CHBA metabolite, (**g**) blank plasma for baclofen-d4, (**h**) plasma spiked with baclofen-d4 (S-isomer 3.2 min and R-isomer 5.2 min, 1000 ng/mL), (**i**) extracted human plasma sample following baclofen administration and spiked with baclofen-d4.

**Table 1 molecules-25-00250-t001:** Summary of MS/MS and chromatography parameters for both methods. APCI: atmospheric pressure chemical ionization, MRM: multiple reaction monitoring.

	**S-Baclofen**	**R-Baclofen**	**S-CHBA**	**R-CHBA**	**S-BAC-d4**	**R-BAC-d4**
MRM transition *m*/*z* (Q1→Q3)	214.10 > 151.05		213.15 > 151.10		218.10 > 155.10	
MS/MS Ionization	APCI (+) ve Mode		APCI (−) ve Mode		APCI (+) ve Mode	
Q1(V)	−30		15		−30	
CE(V)	−19		13		−19	
Q3(V)	−15		13		−16	
**R-and S-Baclofen Separation and Racemic CHBA Method:**						
Column	Crownpak CR(+) 4.00 × 150 mm, 5μ (Part # 27714)					
Guard Column:	Crownpak CR(+) 4.00 × 10 mm, 5μ (Part # 27714)					
Run Time:	11 min					
Mobile Phase	A-0.4% in formic acid water and B: 0.4% formic in acetonitrile					
Flow	1 mL/min, Isocratic (86:16, A:B)					
Retention time	3.5	5.4	8.7		3.5	5.4
**R-and S-CHBA Separation and Racemic Baclofen Method:**						
Column	Chiralcel OJ R-RH 2.1 × 150 mm, 5μ (Part #17794)					
Guard Column:	Phenomex C18					
Run Time:	27 min					
Mobile Phase	A-0.4% in Formic acid water and B: 0.4% Formic in acetonitrile					
Flow	0.1 mL/min, Isocratic (86:16, A:B)					
Retention time	5.0		21.0	23.5	5.1	

**Table 2 molecules-25-00250-t002:** Inter-day and intra-day accuracy (% bias) and precision (%RSD) for baclofen and its metabolite in human plasma. HQC: high-quality control, LLOQ: lower limit of quantification, LQC: low-quality control, MQC: middle-quality control.

Conc. (ng/mL)	*S*-Baclofen				*R*-Baclofen				CHBA (Racemic)			
	LLOQ	LQC	MQC	HQC	LLOQ	LQC	MQC	HQC	LLOQ	LQC	MQC	HQC
Theoretical Conc.	1 ng/mL	5 ng/mL	500 ng/mL	1500 ng/mL	1 ng/mL	5 ng/mL	500 ng/mL	1500 ng/mL	2 ng/mL	10 ng/mL	1000 ng/mL	3000 ng/mL
%Bias_intra-assay_	11.1	0.4	1.7	−1.3	7.4	0.6	−0.9	−3.4	6.0	−5.2	2.9	−3.9
%Bias_inter-assay_	14.1	−1.3	4.0	−2.3	7.9	8.9	0.2	−3.8	18.4	4.1	4.2	−2.3
% RSD^intra-assay^	7.7	2.4	1.9	1.6	2.1	5.2	3.6	1.1	11.0	5.0	3.2	1.2
% RSD^inter-assay^	16.6	4.2	3.9	4.3	5.8	8.1	1.7	2.1	12.1	9.8	5.5	3.3

**Table 3 molecules-25-00250-t003:** Absolute extraction recoveries and absolute matrix effects of the baclofen and its metabolite from human plasma and cerebral spinal fluid (CSF) (Mean ± SD, *n* = 3).

Bio-Matrix	Analytes	Absolute Extraction Recovery (%)			Absolute Matrix Effect (%)		
		LQC	MQC	HQC	LQC	MQC	HQC
Human plasma	S-baclofen	94.2 ± 8	84.8 ± 2.3	85 ± 6.7	94.7 ± 7.4	85.8 ± 3.9	89.7 ± 7.7
	R-baclofen	99.4 ± 7.7	99.2 ± 7.7	88.6 ± 10.6	87.7 ± 6.1	91.4 ± 2.0	94.7 ± 16.3
	Racemic CHBA	99.9 ± 9.3	107.7 ± 7.5	109.1 ± 6.2	96.4 ± 1.7	96.3 ± 4.9	100.2 ± 1.5
CSF: Plasma (1:1)	S-baclofen	82.2 ± 2.4	103.0 ± 10.8	96.8 ± 0.2	83.4 ± 4.3	88.2 ± 5.3	89.9 ± 4.4
	R-baclofen	95.8 ± 2.1	94.7 ± 3.2	97 ± 0.6	84 ± 3.9	86.5 ± 1.0	91.8 ± 5.3
	Racemic CHBA	97.9 ± 9.3	109.2 ± 0.9	101.7 ± 6.4	94.5 ± 10.6	105.2 ± 6.1	101.9 ± 7.9

**Table 4 molecules-25-00250-t004:** Mean stability recoveries of baclofen and its metabolites at different storage conditions in human plasma.

Storage Conditions	Conc.	S-Baclofen		R-Baclofen		CHBA (Racemic)	
		Measured Mean Conc. (ng/mL)	Accuracy (%)	Measured Mean Conc. (ng/mL)	Accuracy (%)	Measured Mean Conc. (ng/mL)	Accuracy (%)
Bench-top stability (20 °C, up to 6 h)	LQC	4.9 ± 0.3	97.3 ± 6.1	5.0 ± 0.5	99.4 ± 9.1	10.7 ± 1.1	106.9 ± 11.0
	HQC	1520.0 ± 32.3	101.3 ± 2.1	1490.1 ± 18.8	99.3 ± 1.3	2771.2 ± 829.0	104.4 ± 2.2
Freeze–thaw stability (−80 °C, up to three cycles)	LQC	5.0 ± 0.3	100.5 ± 6.6	5.3 ± 0.4	105.9 ± 8.1	9.8 ± 0.9	98.1 ± 8.6
	HQC	1551.2 ± 19.9	103.4 ± 1.3	1511.5 ± 10.7	100.8 ± 0.7	3148.8 ± 47.7	105.0 ± 1.6
Long-term stability (−80 °C, 40 days)	LQC	4.6 ± 0.2	92.1 ± 4.2	4.7 ± 0.2	94.3 ± 3.1	10.0 ± 1.4	100.4 ± 13.6
	HQC	1533.1 ± 1.9	102.2 ± 0.1	1473.2 ± 22.4	98.3 ± 1.5	1533.1 ± 28.1	107.0 ± 0.9
Auto-sampler stability (4 °C, 72 h)	LQC	5.3 ± 0.2	105.9 ± 3.6	5.4 ± 0.1	108.6 ± 2.1	5.3 ± 3.1	98.2 ± 0.1
	HQC	1492.8 ± 7.0	99.5 ± 0.4	1501 ± 2.8	100.2 ± 0.2	1492.8 ± 118.0	100.8 ± 4.0
Processed Samples’ long-term stability (−80 °C, 40 days)	LQC	5.3 ± 0.2	105.4 ± 3.5	5.6 ± 0.1	111.8 ± 1.1	5.3 ± 3.0	107.6 ± 0.9

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
