# Peer review of "Simultaneous Quantitation of S(+)- and R(−)-Baclofen and Its Metabolite in Human Plasma and Cerebrospinal Fluid using LC–APCI–MS/MS: An Application for Clinical Studies"

_molecules, 2020, doi:10.3390/molecules25020250_

Round 1
Reviewer 1 Report
The authors describe a chiral separation with LC-MS of a racemic pharmaceutical to monitor its dosing regime. The manuscript is written well and contains all necessary information. There are only a few minor issues.
One is terminology. I do realize that FDA in their documentation relates to accuracy in the sense of trueness. Yet the internationally accepted terminology, like in ISO5725 or the International Vocabulary of Metrology, refers to trueness for any systematic biases and precision for the random variability. Accuracy is then the combination of both. Therefore, I recommend that the authors adjust this accordingly in their manuscript.
The sentence on line 102ff needs revision.
The retention times on line 107 do not correspond to the retention times shown in Tab. 1. Furthermore, the layout of all the tables must be edited. For tables breaking over two pages the column headers need to appear on each page and the alignment in all tables is off.
L 123: Sensitivity relates to the change of signal over change of amount of analyte in analytical chemistry. What is described here are performance limits.
Abbreviations like LLOQ (L124) or LQC, MQC, and HQC (L153) need to be defined before first use.
L 163: Recovery is an ambiguous term. What the authors are describing here is the extraction efficiency of their analytes. For others, recovery might mean how much is found at the end of the whole procedure, so the combination of extraction efficiency and matrix effect in LC-MS. This needs to be clarified.
L 233: "Post-column infusion of 0.5% ammonium hydroxide (NH4OH)..." is not intuitive for the detection of a ammino-function bearing compound in positive ion mode with MS. A few words why the authors believed this to be necessary would help the reader in understanding.
L308: The distinction between "ME" and "IS normalized ME" appears a bit superfluous. Stable-isotope labelled analogues of the analytes are added to, amongst other things, control matrix effects. To then also list matrix effects calculated without the normalization through the isotopologue does not add any valuable information and unnecessarily inflates the manuscript.
L 329: an "a" missing on "plasm."
Author Response
Comments and Suggestions for Authors
Reviewer 1
The authors describe a chiral separation with LC-MS of a racemic pharmaceutical to monitor its dosing regimen. The manuscript is written well and contains all necessary information. There are only a few minor issues.
One is terminology. I do realize that FDA in their documentation relates to accuracy in the sense of trueness. Yet the internationally accepted terminology, like in ISO5725 or the International Vocabulary of Metrology, refers to trueness for any systematic biases and precision for the random variability. Accuracy is then the combination of both. Therefore, I recommend that the authors adjust this accordingly in their manuscript.
Reply: We have modified our terminology to identify that the accuracy is expressed as percent bias (% bias) and that precision is presented as relative standard deviation (%RSD). The inter-day accuracy (%bias) and precision (%RSD) are presented in Table 2. This information has been included in the manuscript on lines 163 to help clarify.
The sentence on line 102ff needs revision.
Reply: We have made the corrections and the sentence has been modified in the manuscript
line102. R and S-CHBA metabolites were separated using a Chiralcel OJ R-RH column with 0.4% FA in water and 0.4% FA in ACN (86:16, v/v) as the mobile phase and a flow rate of 0.1 mL/min. These conditions resulted in acceptable peak shape for R and S-CHBA metabolites with no interference of endogenous compounds at the retention time for both baclofen and its CHBA metabolites (Supplementary Figure 1). Post column infusion of 0.5% ammonium hydroxide (NH4OH) in water-ACN (80:20, v/v) was accomplished using a 3-way connector before mass detection to improve ionization for the R- and S-CHBA metabolites, which utilized negative APCI mode ionization (data not shown) [1].
The retention times on line 107 do not correspond to the retention times shown in Tab. 1. Furthermore, the layout of all the tables must be edited. For tables breaking over two pages the column headers need to appear on each page and the alignment in all tables is off.
Reply: We have corrected this in the manuscript (line 116) and Table 1.
L 123: Sensitivity relates to the change of signal over change of amount of analyte in analytical chemistry. What is described here are performance limits. Sensitivity was well defined as the lower limt of quantitation (LLOQ), where the all analytes signal to noise (S/N) response was equal or greater than 10 fold the blank response.
Reply: We have made corrections, and the sentence has been modified in the manuscript line 148. Sensitivity was defined as the lower limit of quantitation (LLOQ), where the all analytes signal to noise (S/N) response was equal or greater than 10 fold compared to the blank response.
Abbreviations like LLOQ (L124) or LQC, MQC, and HQC (L153) need to be defined before first use.
Reply: We have provided the full name and abbreviation in the manuscript on line166.
L 163: Recovery is an ambiguous term. What the authors are describing here is the extraction efficiency of their analytes. For others, recovery might mean how much is found at the end of the whole procedure, so the combination of extraction efficiency and matrix effect in LC-MS. This needs to be clarified.
Reply: We have clarified this terminology in the manuscript. The absolute recoveries have evaluated as specified by Matuszewski et al-2003.
Reference:B.K. Matuszewski, M.L. Constanzer, C.M. Chavez-Eng, Strategies for the assessment of matrix effect in quantitative bioanalytical methods based on HPLC-MS/MS, Anal Chem 75(13) (2003) 3019-30.
We have added the following at line 204
The absolute recoveries were calculated at the LQC, MQC and HQC concentration as the mean peak area of an analyte spiked before extraction to the mean peak area of an analyte spiked post extraction multiplied by 100.
L 233: "Post-column infusion of 0.5% ammonium hydroxide (NH4OH)..." is not intuitive for the detection of an amino-function bearing compound in positive ion mode with MS. A few words why the authors believed this to be necessary would help the reader in understanding.
Reply: We have modified the sentence in the manuscript in section Chromatographic and mass spectrometric optimization line 108 and added the following reference.
Post column infusion of 0.5% ammonium hydroxide (NH4OH) in water-ACN (80:20, v/v) was accomplished using a 3-way connector before mass detection to improve ionization for the R- and S-CHBA metabolites, which utilized negative APCI mode ionization (data not shown) [1].
Refence: Sanchez-Ponce, R.; Wang, L. Q.; Lu, W.; von Hehn, J.; Cherubini, M.; Rush, R., Metabolic and Pharmacokinetic Differentiation of STX209 and Racemic Baclofen in Humans. Metabolites 2012, 2, (3), 596-613
L308: The distinction between "ME" and "IS normalized ME" appears a bit superfluous. Stable-isotope labelled analogues of the analytes are added to, amongst other things, control matrix effects. To then also list matrix effects calculated without the normalization through the isotopologue does not add any valuable information and unnecessarily inflates the manuscript.
Reply: Ion suppression or ion enhancement due to matrix at the retention times of analytes is a common problem observed in small molecule bioanalysis. In order to avoid matrix effect at the retention time of analytes, the chromatographic method was optimized with the qualitative assessment of matrix effect via mean peak area or peak area (IS normalized ME) were calculated. ME or IS normalized ME values for all analytes were < ±15%, indicating a no matrix effect for the study samples with reproducibility over the all QCs range.
This has been clarified on lines 204 to 206 and line 410.
L 329: an "a" missing on "plasm."
Reply: We have corrected this in manuscript on line 329.
Reviewer 2 Report
the manuscript of He et al is interesting and well written, there are only some minor concerns worthy of investigation:
the first one is that there are several other methods in liquid chromatography tandem mass spectrometry for the quantification of baclofen and its metabolite and not all of them use SPE extraction, so it would be useful to better explain what this method adds in comparison to the others. the authors have fully validated the method but they did not take into consideration a very important parameter which is the "uncertainty" of the measure. Please include also this validation parameter explaining in detail how it was calculated. I suggest to read the following paper of Peters et al. Method Development in Forensic Toxicology. Curr Pharm Des. 2017;23(36):5455-5467.
Author Response
The manuscript of He et al is interesting and well written, there are only some minor concerns worthy of investigation:
The first one is that there are several other methods in liquid chromatography tandem mass spectrometry for the quantification of baclofen and its metabolite and not all of them use SPE extraction, so it would be useful to better explain what this method adds in comparison to the others.
Reply: Thank you for bringing this to our attention. We have made the appropriate corrections, and the sentence was modified in the manuscript in section Chromatographic and mass spectrometric optimization line 118.
Previous studies extracting baclofen from biomatrices utilized different extraction methods including solid phase extraction (SPE) [1, 25, 26], liquid-liquid extraction[27] and protein precipitation extraction(PPE)[1]. However, SPE [25] in our hands resulted in poor CHBA recovery, with only approximately 5% of the metabolite recovered (data not shown). In order to simplify procedures and improve metabolite recovery, we utilized a PPE method to achieve approximately 90% absolute recovery for baclofen and CHBA metabolites (Table 4).
the authors have fully validated the method but they did not take into consideration a very important parameter which is the "uncertainty" of the measure. Please include also this validation parameter explaining in detail how it was calculated. I suggest to read the following paper of Peters et al. Method Development in Forensic Toxicology. Curr Pharm Des. 2017;23(36):5455-5467.
Reply: We have included a sentence describing the utility of the method as requested by reviewer 2 on lines 144 to 146.
The method is quantitative, validated for both plasma and CSF samples and linear from 1 to 1500 ng/mL for baclofen. Our validated method is accurate, precise, and sensitive allowing for the routine analysis of baclofen and metabolites in clinical samples